# The Effects of Temperature Management on Brain Microcirculation, Oxygenation and Metabolism

**DOI:** 10.3390/brainsci12101422

**Published:** 2022-10-21

**Authors:** Katia Donadello, Fuhong Su, Filippo Annoni, Sabino Scolletta, Xinrong He, Lorenzo Peluso, Leonardo Gottin, Enrico Polati, Jacques Creteur, Olivier De Witte, Jean-Louis Vincent, Daniel De Backer, Fabio Silvio Taccone

**Affiliations:** 1Department of Intensive Care, Erasme Hospital, Free University of Brussels, Route de Lennik 808, 1070 Brussels, Belgium; 2Department of Anesthesia and Intensive Care B, Department of Surgery, Dentistry, Gynaecology and Paediatrics, University of Verona, AOUI-University Hospital Integrated Trust of Verona, Policlinico G.B. Rossi, Piazzale Ludovico Scuro, 37134 Verona, Italy; 3Service of Intensive and Critical Care Medicine, Department of Medical Science, Surgery and Neuroscience, University of Siena, 53100 Siena, Italy; 4Department of Intensive Care Medicine, Sun Yat-sen University Cancer Center, Guangzhou 510060, China; 5Departement of Cardio-Thoracic Anesthesia and Intensive Care, Department of Surgery, Dentistry, Gynaecology and Paediatrics, University of Verona, AOUI-University Hospital Integrated Trust of Verona, Piazzale Aristide Stefani, 37100 Verona, Italy; 6Department of Neurosurgery, Erasme Hospital, Free University of Brussels, Route de Lennik 808, 1070 Brussels, Belgium; 7Department of Intensive Care, CHIREC, 1420 Braine L’Alleud, Belgium

**Keywords:** microvascular flow, cerebral oxygenation, hypothermia, brain metabolism, lactate

## Abstract

Purpose: Target temperature management (TTM) is often used in patients after cardiac arrest, but the effects of cooling on cerebral microcirculation, oxygenation and metabolism are poorly understood. We studied the time course of these variables in a healthy swine model.Methods: Fifteen invasively monitored, mechanically ventilated pigs were allocated to sham procedure (normothermia, NT; *n* = 5), cooling (hypothermia, HT, *n* = 5) or cooling with controlled oxygenation (HT-Oxy, *n* = 5). Cooling was induced by cold intravenous saline infusion, ice packs and nasal cooling to achieve a body temperature of 33–35 °C. After 6 h, animals were rewarmed to baseline temperature (within 5 h). The cerebral microvascular network was evaluated (at baseline and 2, 7 and 12 h thereafter) using sidestream dark-field (SDF) video-microscopy. Cerebral blood flow (laser Doppler MNP100XP, Oxyflow, Oxford Optronix, Oxford, UK), oxygenation (PbtO_2_, Licox catheter, Integra Lifesciences, USA) and lactate/pyruvate ratio (LPR) using brain microdialysis (CMA, Stockholm, Sweden) were measured hourly. Results: In HT animals, cerebral functional capillary density (FCD) and proportion of small-perfused vessels (PSPV) significantly decreased over time during the cooling phase; concomitantly, PbtO_2_ increased and LPR decreased. After rewarming, all microcirculatory variables returned to normal values, except LPR, which increased during the rewarming phase in the two groups subjected to HT when compared to the group maintained at normothermia. Conclusions: In healthy animals, TTM can be associated with alterations in cerebral microcirculation during cooling and altered metabolism at rewarming.

## 1. Introduction

Targeted temperature management (TTM) is often used for brain protection after cardiac arrest (CA) or after traumatic brain injury [1,2,3]. Despite improved knowledge of the physiological and pathological changes associated with temperature variations and cooling procedures [4,5,6], concerns have been raised about the balance between neuroprotective mechanisms and potential harm related to hemodynamic impairment, coagulation disorders and arrhythmias. The optimal TTM dose is still debated [7,8,9,10,11]. Hence, further investigations are needed to better evaluate the unwanted effects of cooling procedures [12,13].

In 1962, Ashton et al. showed that temperature influences vascular waterfalls and possibly microvascular flow [14]; hypothermia predisposes to myocardial ischemia, coagulopathy, bleeding, metabolic disorders, immunodepression and sepsis [15,16]. Temperature elevation enhances immune function, antibody production, T-cell activation, neutrophil and macrophage function [17,18]. In particular, TTM with body temperature < 34 °C may contribute to peripheral vasoconstriction and potentially affect microcirculation [19]. Microcirculation is a major determinant of oxygen and nutrient delivery to the tissues, and impaired microcirculatory flow and reactivity have been described in various critical conditions, including cardiogenic or hemorrhagic shock and sepsis [19,20,21]. In these conditions, the cerebral, as well as the sublingual microcirculation, is impaired, with a significant decrease in tissue oxygenation and altered metabolism [22,23]. In an ovine model, He et al. showed that the use of TTM in healthy animals was associated with a significant reduction in sublingual capillary density and increased lactate levels in the absence of major changes in systemic hemodynamics [24]. However, whether this was an adaptive or pathological phenomenon remains unknown, as isolated lactate levels measurement might be due to other factors than alterations in cellular metabolism [25,26]; concurrent pyruvate measurements can more precisely evaluate the consequences of impaired tissue perfusion on local metabolism [27].

In TTM after CA, experimental evidence remains controversial. Additionally, favorable results on brain cell survival were published on an experimental model of hypothermia after ischemic and traumatic injury in rats and mice [28,29,30]; a few studies reported a positive impact of mild HT on cerebral microcirculation in rats [31,32], and others reported a decrease in pigs [33]. In particular, TTM could reduce microvascular cell apoptosis after a reperfusion injury and therefore protect from microcirculatory abnormalities [34]. However, whether these effects could also be observed in healthy animals remains unknown. Last, as hypothermia reduces cell metabolism, it is unknown whether systemic and brain microcirculation may decrease accordingly as an adaptative phenomenon or if they might be affected by the reduced temperature, leading to a discrepancy between flow and cell metabolism.

The aim of this study was, therefore, to evaluate the effects of TTM on brain microcirculation, oxygenation and metabolism in a healthy pig model.

## 2. Materials and Methods

### 2.1. Study Approval

The study protocol was approved by the Institutional Review Board for Animal Care of the Free University of Brussels (Brussels, Belgium; Acceptance Number 404N). Care and handling of the animals were in agreement with the “Guide for Care and Use of Laboratory Animals” published by the U.S. National Institute of Health (NIH Publication eight edition, update 2011) and the National Institutes of Health Guidelines (Institute of Laboratory Animal Resources).

### 2.2. Experimental Animals

Fifteen domestic pigs (*Sus Scropha)*, weighing between 36 and 44 kgs, were fasted in a ventilated and controlled temperature chamber (20–24 °C) for 24 h before the study, with free access to water. On the day of the experiment, the animals were premedicated with intramuscular midazolam (0.25 mg/kg, Dormicum, Roche SA, Beerse, Belgium) and ketamine hydrochloride (20 mg/kg, Imalgine, Merial, Lyon, France) and placed in the supine position on an operating table. A superficial ear vein was cannulated with a peripheral venous 20-gauge catheter (Surflo IV Catheter, Terumo Medical Company, Leuven, Belgium). An intravenous administration of 30 mcg/kg fentanyl citrate (Janssen, Beerse, Belgium), 5 mg/kg thiopental (B-Braun Medical SA, Rubi, Barcelona) and 0.5 mg/kg rocuronium bromide (Esmeron, Organon, Oss, The Netherlands) was used for endotracheal intubation (6.5 mm endotracheal tube, Hi-Contour, Mallinckrodt Medical, Athlone, Ireland). Thereafter, all pigs were anesthetized with a continuous intravenous administration of thiopental (3 mg/kg·h), ketamine hydrochloride (0.5 mg/kg·h) and morphine (0.2 mg/kg·h). In order to avoid movement artifacts, a muscular blockade was achieved with a continuous infusion of 10 mcg/kg·h of rocuronium throughout the experiment. Boluses of fentanyl (15 mcg/kg) and of thiopental (2 mg/kg) were administered if needed in case of tachycardia and/or hypertension, suggesting insufficient anesthesia; boluses of rocuronium (0.5 mg/kg) were given to avoid shivering. Mechanical ventilation (Servo 900 C ventilator; Siemens-Elema, Solna, Sweden) was initiated with a tidal volume of 10 mL/kg, respiratory rate of 12–16 breaths/minute, positive end-expiratory pressure (PEEP) of 5 cm H_2_O, an inspired oxygen fraction of 1 (FiO_2_), inspiratory time to expiratory time ratio (I/E) of 1:2 and a square-wave pattern. Respiratory rate was adjusted to maintain end-tidal carbon dioxide pressure (PetCO_2_, 47,210 A Capnometer; Hewlett Packard GmbH, Boehlingen, Germany) between 35 and 45 mmHg before arterial cannulation and blood gas analyses were available.

### 2.3. Surgical Procedures

The right femoral artery and vein were surgically exposed. A 6F arterial catheter (Vygon, Cirencester, UK) was invasively introduced into the femoral artery; an introducer was inserted through the femoral vein, and a 7F pulmonary artery (PA) catheter (Edwards Life Science, Baxter, Irvine, CA, USA) was advanced under the monitoring of pressure waveforms and connected to pressure transducers (Edwards Lifescience, Irvine, CA, USA), zeroed at mid-chest level. Core temperature, cardiac output and PA pressures were measured by Sirecust 404 (Siemens, Germany) and Vigilance monitor (Edwards Lifesciences Corporation, Irvine, CA 92614, USA). Both catheters were flushed intermittently with a heparinized solution (2 UI/mL heparin sodium in 500 mL 0.9% saline solution). A small midline laparotomy was performed to expose the bladder, and a 14F Foley catheter (Beiersdorf AG, Hamburg, Germany) was placed to record urinary output throughout the experiment. The abdomen was then closed in two layers by end-knot continuous sutures. During the surgical operation, Ringer’s lactate and 6% hydroxyethyl starch (HES) solutions were titrated to prevent hypovolemia (infused at rates of 2 mL/kg·h and 1 mL/kg·h, respectively). After vascular and abdominal surgery, the animals were turned to the prone position.

### 2.4. Cerebral Surgery

The scalp was opened with a cross-shape incision using an electric bistoury [14]. The skin was then sutured to maintain full access to the head bones. Two large bilateral craniotomies were then performed in all animals using a high-speed drill, and a fine wire saw (Aesculap-Werke AG, Tuttlingen, Germany) to open two holes of about 3 × 3 cm on the left and 2 × 1.5 cm on the right side of pig frontoparietal bones. The dura covering the left fronto-parietal lobe was then opened with a large incision, carefully avoiding any cortical damage and exposing the brain. Bleeding from the skull was controlled using surgical wax and blood, or cerebrospinal fluid was gently removed by saline solutions without any direct contact with the brain. Left craniotomy was eventually used to assess brain microcirculation. On the right side, dura mater was punctured to insert a microdialysis (MD) catheter (CMA, 20; cut-off membrane, 20 kDa; membrane length, 10 mm; CMA Microdialysis AB, Stockholm, Sweden), a Clark electrode (Licox catheter, Integra Lifesciences, Princeton, NJ 08540, USA) for tissue oxygen pressure (PbtO_2_) measurement and a catheter to measure brain temperature (Licox catheter, Integra Lifesciences, Princeton, NJ 08540, USA). All catheters were placed under sterile conditions at a depth of 0.8–1.0 cm into the brain parenchyma so that their tips were located in the frontal grey matter. Finally, catheters were sutured to the skin to avoid displacement during the experiment (the correct positioning of the catheters was further confirmed in *post-mortem* examination). A third tiny hole was eventually performed through the skull with a needle-like drill so as to insert a laser-doppler catheter (laser Doppler MNP100XP, Oxyflow, Oxford Optronix, Oxford, UK); its right position catheter was confirmed by the presence of values at least 800 Units; the catheter was then fixed with wax to the skull to prevent any additional movements. The craniotomy holes were then protected by wet sterile gauzes, avoiding any contact with the brain cortex. Brain desiccation was prevented by local hourly administration of 1–2 mL saline solution. All animals were then allowed to stabilize before baseline measurements were recorded.

### 2.5. Monitoring and Measurements

Arterial and mixed venous blood samples were analyzed hourly with automated analyzers (ABL725 and OSM3, Radiometer Medical A/S, Akandevey 21, DK-2700 Brønshøj, Denmark). Hemoglobin concentration and oxygen saturation were measured with an analyzer calibrated for animals (OMS3; Radiometer). All monitored variables were recorded every 60 min. Measurements of mean arterial pressure (MAP), pulmonary arterial pressure (PAP), right atrial pressure (RAP) and pulmonary occlusion arterial pressure (PAOP) were obtained at the end of expiration (Sirecust 404; Siemens, Erlangen, Germany). Core temperature and cardiac output (Vigilance; Baxter, Edwards Critical Care) were continuously monitored. Body surface area [35], cardiac index (CI), stroke volume (SV), systemic vascular resistance (SVR), pulmonary vascular resistance (PVR), left ventricular stroke work index (LVSWI), oxygen delivery (DO_2_), oxygen consumption (VO_2_) and oxygen extraction (OER) were calculated using standard formulas. The total amount of blood withdrawn for analyses was less than 50 mL (i.e., around 2% of the estimated total blood volume of each animal).

### 2.6. Fluid Management

Ringer’s lactate solution (RL) and 6% hydroxyethyl starch solution (HES, Voluven, Fresenius Kabi, Belgium) were initially infused via the cephalic vein catheter at a rate of 2 and 1 mL/kg·hour, respectively; fluid administration was then titrated to maintain PAOP similar to baseline values. Potassium chloride and glucose were administered in case of hypokalaemia (<3.5 mEq/L) and hypoglycaemia (<40 mg/dL), respectively.

### 2.7. Cerebral Microcirculation, Temperature, Oxygen and Metabolism Assessment

The microvascular network of the cerebral cortex was visualized using an SDF videomicroscopy system (MicroScan^TM^, MicroVisionMedical Inc., Amsterdam, The Netherlands) with a 5× imaging objective living 326× magnification. The lens of the imaging device was covered with a disposable sterile cap and was applied without pressure to the cerebral frontoparietal cortex. Because of brain pulsatility, this was best accomplished by placing the animal head and the device on a metallic arm for stabilization (MAYFIELD Triad^TM^ Skull Clamp A1108, Integra SM, Benelux). The absence of external superimposed pressure was confirmed by the preservation of flow in large vessels [36]. After each set of video captures, the SDF device was removed, and the brain cortex was protected with sterile saline-soaked gauze. Guided by previously published studies in similar models [22,23,24], images were recorded at 4 different time points: at baseline (T0), and then 2 (T2), 7 (T7) and 12 (T12) hours thereafter. At least five videostrips from different areas, each of minimum 20 s, were recorded on a disk using a computer and a video card (Micro Video; Pinnacle Systems, Mountain View, CA, USA). The images were then stored under a random number for further analysis. An investigator blinded to group allocation and time later analyzed these sequences semi-quantitatively offline [36,37]. Vessel size was determined using a micrometer scale, and the vessels were separated into large and small vessels, using a diameter cut-off value of 20 μm [36,38]. Microcirculatory analysis was based on the principle that density of the vessels is proportional to the number of vessels crossing arbitrary lines (three equidistant vertical lines and three equidistant horizontal ones drawn on the screen). Vessel density was calculated as the number of vessels crossing the lines divided by the total length of the lines; the proportion of small-perfused vessels (PSPV) can be calculated using the total amount of counted vessels and those which present intermittent flow and those without flow; the functional capillary density (FCD) was calculated as the product of vascular density and perfused capillary density. The mean flow index (MFI) is a score based on the determination of the predominant type of flow (absent, intermittent, sluggish or normal) in four quadrants dividing the screen. As perfusion heterogeneity represents a crucial determinant of the extraction capabilities of the tissues, with the homogeneous flow being better tolerated even if blood flow is reduced, we calculated a heterogeneity index [36] for MFI, PPV and FCD, defined as the difference between maximal and minimal values, evaluated at each point in the five areas divided by the mean value of the five areas. In each animal, the data from the investigated areas were averaged for each time point. The intracranial temperature and PbtO_2_ catheters were connected to specific monitoring (Brain Temperature Monitoring and Brain Tissue Oxygen Monitoring, AC31, Integra Lifesciences, Princeton, NJ 08540, USA), where the initial temperature was manually adjusted to blood temperature, and data were collected hourly. The probe function was confirmed by an oxygen challenge (FiO_2_ 1.0 for 2 min). In order to allow probe equilibration, data from the first hour after placement were not used. The CMA 20 catheter was perfused with CNS perfusion fluid (148 mM NaCl, 2.7 mM KCl, 1.2 mM CaCl_2_ and 0.85 mM MgCl_2_; Osmolality 305 mOsm/kg; pH 6) at a flow rate of 1.0 μL/min by a miniaturized infusion pump (CMA 107, CMA Microdialysis AB, Solna, Sweden). This flow rate guaranteed almost 30–50% recovery rate for molecules of less than 20 kDa and was selected to provide additional fluid for further research on brain metabolites. After one hour of stabilization, the perfusate was collected every 60 min in specific microvials. Samples were analyzed for lactate, pyruvate, glycerol, glutamate and glucose by an automatic analyzer (CMA 600 Microdialysis Analyzer, CMA Microdialysis AB, Stockholm, Sweden). The lactate/pyruvate ratio (LPR) was automatically calculated.

### 2.8. Experimental Protocol

After the surgical procedures, baseline measurements, including cerebral microcirculation and cerebral dynamic data collection, were obtained. Animals were randomly allocated to groups according to pre-sealed envelopes. At first, the study was conceived to compare normothermia (NT) with induced hypothermia (HT). Nevertheless, we observed after the first animal submitted to HT a significant increase in brain and systemic oxygenation (PbO_2_ and PaO_2_, in temperature-adjusted blood gas values), which may have affected our final results. Thus, after approval of the protocol modification from the animal Ethics Committee, we added a third study group, which underwent HT with PaO_2_ concentrations kept constant (HT-Oxy) through repeated initial blood gas analyses (i.e., every 30 min until stable hypothermia was reached). In the absence of previous studies to calculate the effects of HT on brain microcirculation, we estimated a convenient sample size of 5 animals for each group, and the study protocol was designed on another experimental study evaluating mild hypothermia performed in our institution [24]. Animals included in the NT were kept at constant baseline temperature (i.e., around 39 °C) over the study period using ice packs or an electrothermal blanket whenever needed; after baseline measurements (T0), a bolus of 30 mL/Kg (at a rate of 1000 mL/h) of ambient temperature NaCl 0.9% solution was given. No other interventions were performed. The animals allocated to the two HT groups received a 30 mL/Kg of NaCl 0.9% solution stored at 4 °C saline infusion at the same infusion rate as the NT group. Moreover, ice packs were simultaneously placed around the animal, and a nasal cooling set (RhinoChill System, Benechill, CA, USA) was applied by insufflating the mixed vapor of a perfluorochemical and air into the pigs’ rhinopharynx tract at a flow of 40–60 L/min for one hour. The targeted core temperature (continuously measured via the PA catheter) was to be reached as soon as possible and maintained around 34 °C for 6 h, by applying or removing ice packs. After this period phase (T7), all ice packs were removed, and the animals were actively rewarmed by an electrothermal blanket to baseline temperature over 5 h (around 1 °C/h). In animals allocated to the HT-Oxy group, ventilator FiO_2_ was adjusted according to the temperature-corrected blood gas values to maintain a PaO_2_ at baseline values during the entire study period. All variables were recorded hourly over the study period. At the end of the study, the animals were euthanized using the injection of a high dose of intravenous potassium chloride.

### 2.9. Statistical Analysis

Statistical analysis was performed using SPSS 24.0 for Windows (SPSS Inc, Chicago, IL, USA). Data are presented as mean ± SD or median (range). Normal distribution was confirmed with the Kolmogorov–Smirnov test. Variables were compared with parametric Student’s t-test or the Mann–Whitney U test for non-parametric data. The significance of differences in the measured variables between groups was analyzed using a two-way (time and groups) analysis of variance for repeated measures, followed by a Bonferroni post hoc analysis. In order to estimate the correlation (expressed as “r” coefficient) between different variables in the presence of repeated measurements, we used a mixed model with the SAS system (version 9.2, SAS Institute Inc., NC 27513, USA). A *p*-value < 0.05 was considered statistically significant.

## 3. Results

At baseline, there were no significant differences between groups in hemodynamic and respiratory variables, cerebral microcirculation, oxygenation and metabolic data (Appendix A).

During the cooling phase, core and brain temperatures remained stable in the NT group, while they decreased significantly in the HT and HT-Oxy groups (Appendix A, Figure 1).

In particular, body temperature decreased below 35 °C within the first 2 h, while brain temperature was around 32 °C within the same time period (Figure 1).

In the NT group, all variables remained stable throughout the study period. After an initial increase in MAP, SV and CI related to cold fluids administration to reduce body temperature, there was a significant decrease in CI in the HT and HT-Oxy groups (Appendix A); PAOP and RAP increased only in the initial phase in the two groups. In both HT and HT-Oxy groups, heart rate (HR), CI and MPAP returned to baseline levels after the rewarming phase, while MAP remained lower than baseline at the end of the experimental protocol. Serum lactate slightly decreased in all groups during the study period, while SvO_2_ remained stable in the NT group and increased in the two others (Appendix A; Figure 1); DO_2_ and VO_2_ decreased during the cooling phase in the HT and HT-Oxy groups and recovered at even higher than baseline values during rewarming. In HT and HT-Oxy groups, both SVRI and PVRI increased during the maintenance phase when compared to NT, while decreased during rewarming to lower values than baseline for PVRI in the HT-Oxy group and for SVRI in both cooled groups (Appendix A).

In the HT and HT-Oxy groups, cerebral FCD, PPV and MFI decreased during the cooling phase and returned to baseline during rewarming, partially in the HT group and completely in the HT-Oxy group (Figure 2, Appendix A) when compared to the NT group.

Despite not being statistically significant, a higher heterogeneity index for all three microvascular variables was observed in both HT and HT-Oxy groups during the cooling period when compared to the NT group. Cerebral blood flow significantly decreased during the cooling phase in the HT and HT-Oxy groups and completely recovered at rewarming (Appendix A).

Cerebral oxygenation increased progressively during cooling only in the HT group, in conjunction with a significant increase in PaO_2_, while it remained close to baseline values in the HT-Oxy and NT groups (Figure 3, Appendix A).

In all animals, cerebral glucose levels decreased progressively together with plasma glucose levels (Appendix A) as well as cerebral glycerol levels, while cerebral glutamate levels remained stable in all groups (Figure 3, Appendix A). The LPR decreased in the HT and HT-Oxy groups during the cooling phase, with a progressive increase above baseline values during the rewarming phase, in particular in the HT group (Appendix A and Figure 3; *p* < 0.05 vs. NT group).

## 4. Discussion

The key findings of this experimental study are that cortical microvascular flow is progressively reduced during cooling, in particular when HT is associated with increased systemic oxygenation. The reduction in cerebral capillary perfusion was associated with stable or increased oxygen concentrations and preserved metabolism during cooling. During rewarming, altered cerebral metabolism persisted but with different patterns that could be related to different tissular oxygen concentrations.

Microcirculation may play a crucial role when cellular metabolism adapts to temperature variations: in particular, when the metabolic rate slows down, the precapillary resistance vessels constrict to reduce blood flow, resulting in less effective microcirculation perfusion capability. This is indicated by a reduction in FCD, PPV and MFI during hypothermia [39]. In swine undergoing CA, Wu et al. reported a similar decrease in MFI during hypothermia [33]. Moreover, in a previous study, we reported that mild HT was associated with a decrease in MFI when compared to normothermia, but all animals developed global hemodynamic alterations [24]. Whether the differences with our present observations are due to differences in species, cooling technique or the presence of some reperfusion injury remains to be determined. However, unlike the effects of microcirculatory abnormalities on tissue perfusion and organ function in critically ill patients [40,41], hypothermia-associated microvascular changes in healthy animals neither resulted in increased cerebral lactate levels nor in cerebral hypoxia, as previously demonstrated for the whole body [24,42]. Although this might be interpreted as a “physiological response”, the increase in the HI during the cooling phase, as well as the partial restoration of brain microvascular density and the increase in LPR at rewarming, suggest a pathological process. Indeed, if the reduction in microvascular flow was expected at low body temperature, a global reduction in capillary density and flow would have resulted in a low HI and have allowed a return of PPV and MFI to baseline. Whether these effects are due to hypothermia itself, the rewarming rate or the concomitant hyperoxia remains to be further elucidated.

Cerebral effects of cooling, when associated with hyperoxia, have never been specifically addressed in the literature. Despite its known adverse effects [43], normobaric hyperoxia (NH) has been suggested as a potential therapy in some specific conditions, such as sepsis [44] and ischemia-reperfusion injury [45]. Nevertheless, in agreement with animal data showing a reduction in muscular flow associated with high oxygen exposure [42,43,44,45,46,47,48], NH was also associated with detrimental effects on the peripheral microcirculation (i.e., sublingual mucosa and thenar eminence) of healthy volunteers [49]. Normobaric hyperoxia might also negatively impact cerebral activity and cognition [50]. Indeed, a recent meta-analysis of 25 randomized controlled trials on critically ill patients showed that a liberal oxygen strategy was associated with increased mortality without any beneficial effects on organ function [51]. Similar results were observed in a subgroup of cardiac arrest survivors [52]. Even though the recent COMACARE study found no association between cerebral oxygenation measured with NIRS and NSE concentrations and outcome in patients resuscitated from OHCA exposed to normoxia or NH [53], a recent meta-analysis found that severe hyperoxemia after CA is associated with worse neurological outcome and increased mortality [54]. Whether the clinical effects of NH on post-anoxic injury are related to the detrimental effects of hyperoxia on the reperfusion injury or are partially enhanced by the concomitant use of hypothermia needs to be further evaluated. Moreover, PaO_2_ values exceeded 160 mmHg in the three groups over time, so it remains unclear whether the differences in PaO_2_ in our study might explain the potentially detrimental effects of oxygen.

Since the beginning of the cooling process, we were able to identify microcirculatory abnormalities in both groups treated with hypothermia. Interestingly, those alterations were not associated with significant changes in MAP or CI, despite changes in systemic blood pressure could be recorded among groups. On the one hand, those findings are consistent with previous studies in clinical and experimental settings, in which similar microcirculatory alterations were described in the brain and sublingual area, despite adequate global hemodynamics [22,23,24,37]. These data suggest that even when severe hypotension is avoided, microcirculatory alterations may occur, supporting the idea of a net dissociation between the micro and the macrocirculation. On the other hand, we cannot exclude that BP changes might have participated to the microcirculatory alterations that were recorded, with the brain not being naive but under the effects of both anesthesia and surgical procedures.

These microcirculatory alterations were not associated with significant increases in blood lactate concentrations. Neurochemical disturbances can also be assessed by microdialysis, which typically includes elevated glutamate (excitotoxicity), glycerol (degradation of cellular membrane) and LPR (i.e., mitochondrial dysfunction, hypoxia, neuroglycopenia), as well as low glucose concentrations (reflecting systemic hypoglycemia and/or low regional flow) [55,56]. Augmentation of LPR was associated with poor prognosis in different settings, such as OHCA, subarachnoid hemorrhage and traumatic brain injury [57,58,59,60]. In pigs resuscitated after cardiac arrest, mild HT was associated with a significant reduction in secondary energy failure, as suggested by a delayed increase in LPR in normothermic animals [61,62,63,64]. In our study, brain lactate was relatively high, with borderline LPR at baseline, in all three groups, and we cannot exclude a possible impact of surgery; nevertheless, hypothermia was associated with a reduction in LPR, suggesting a reduction in cerebral metabolism. LPR showed a subsequent increase during rewarming, while the other metabolites did not differ between groups. Thus, during cooling, decreased perfusion in the microcirculation was not associated with significant metabolic changes within the oxygen pathway. However, in the HT and HT-Oxy groups, a pronounced increase in LPR at rewarming, along with normal brain oxygenation, decreased cPyr and cGlu and a contemporary increase in cLac levels, suggested the occurrence of an hyperglycolytic metabolism [65,66]. Differences in cPyr observed between the HT and HT-Oxy groups might be related to hyperoxia, as previously described in TBI patients [67]. In order to determine whether the application of HT would influence glycolysis and oxidation of lactate, more direct measurements of the cerebral metabolic rate and levels of NADH/NAD^+^ should be performed.

There are some limitations to this study. First, we studied healthy animals, and no conclusions on brain injury situations can be drawn; this study is part of a wider project that we have been conducting with the aim of evaluating brain dynamics during and after cardiac arrest, along with various therapeutic options, mild hypothermia included. This study represents the beginning of our project, the groups of healthy animals to be compared with ill/injured ones; this choice limits the applicability of our results to other areas of research on brain damage. Second, since full anesthesia was necessary, no clinical correlation between reduced microvascular blood flow and clinical abnormalities could be established. Third, any potential effects on microcirculation due to changes in intracranial pressure were prevented by the chosen surgical technique. Fourth, we analyzed limited regions of the brain microvasculature that could not be representative of deeper structures, such as the white matter or the brain stem. Fifth, we did not analyze the release of oxygen species nor anatomo-pathological correlates, limiting the interpretability of our findings. We also could not assess whether microcirculatory disturbances would have any impact on brain function, as evaluated by electroencephalography or neuropathologic examination of the frontal lobe. Moreover, no mechanistic explanations of the observed phenomena were provided. Sixth, the cerebral temperature was lower than the core, probably related to the use of the Rhinochill system and to the craniotomy, and this aspect does not enable us to translate our results into the clinical application of TH. Finally, the study timeline, despite being in line with the available experimental literature, is shorter than the ones applied in clinics, and rewarming was performed at a 1 °C/h speed (as per protocol design); this could have underestimated the risks related to both cooling and rewarming procedures. We cannot exclude that the excess in both cooling depth and rewarming rate might have partially been responsible for the observed phenomena.

## 5. Conclusions

In healthy animals treated with TTM, cerebral microvascular perfusion was reduced during cooling, and the metabolism was altered at rewarming, potentially enhanced by concomitant hyperoxia.

## Figures and Tables

**Figure 1 brainsci-12-01422-f001:**
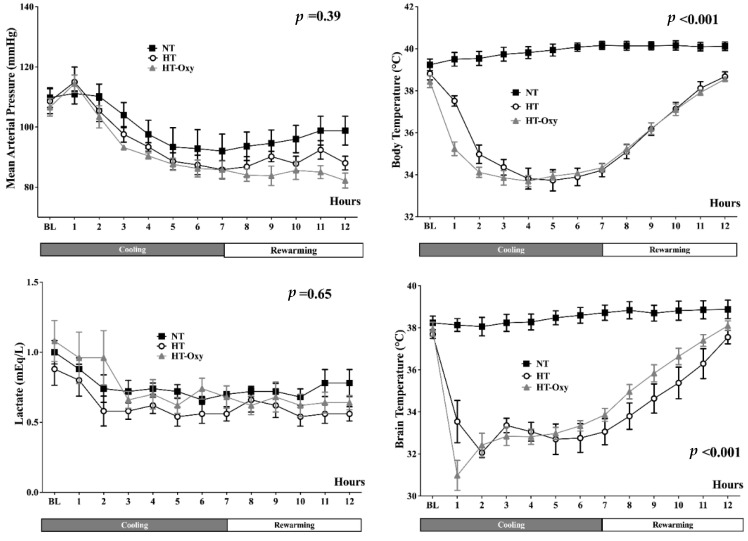
Time-course of mean arterial pressure, lactate concentrations, body and brain temperatures in the normothermia (NT), hypothermia (HT) and hypothermia with PaO_2_ concentrations kept constant with baseline values (HT-Oxy). *p*-value indicated differences over time between groups.

**Figure 2 brainsci-12-01422-f002:**
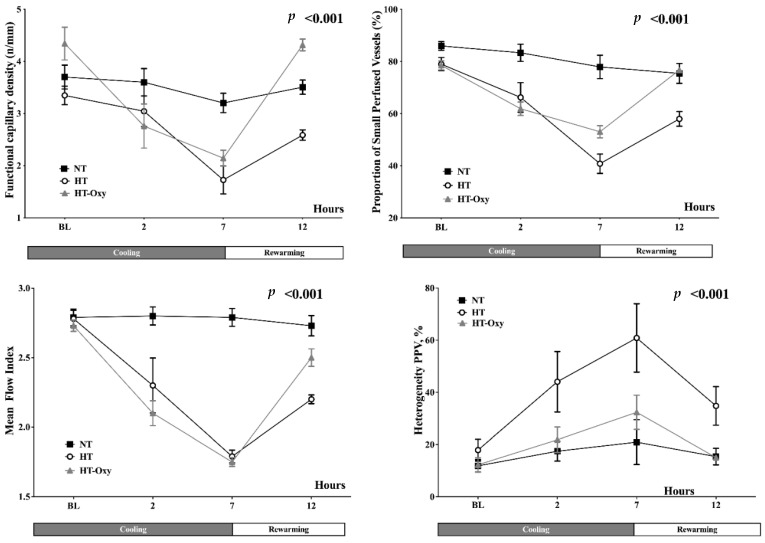
Time-course of microvascular parameters, including the heterogeneity of the proportion of small perfused vessels (PPV) in the normothermia (NT), hypothermia (HT) and hypothermia with PaO_2_ levels kept constant with baseline values (HT-Oxy). *p*-value indicated differences over time between groups.

**Figure 3 brainsci-12-01422-f003:**
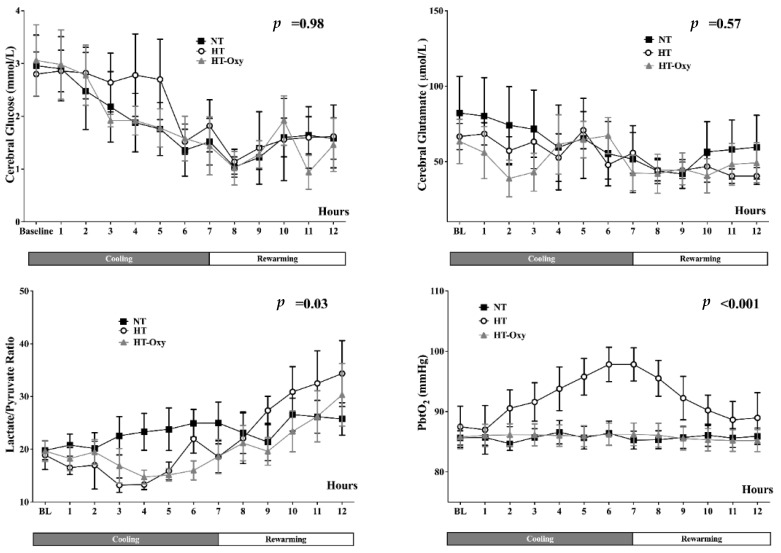
Time-course of cerebral glucose, glutamate, lactate/pyruvate ratio and brain oxygen pressure (PbtO_2_) in the normothermia (NT), hypothermia (HT) and hypothermia with PaO_2_ levels kept constant with baseline values (HT-Oxy). *p*-value indicated differences over time between groups.

## Data Availability

Available under request to the corresponding author.

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
