# Peer review of "The Effects of Temperature Management on Brain Microcirculation, Oxygenation and Metabolism"

_brainsci, 2022, doi:10.3390/brainsci12101422_

Round 1

Reviewer 1 Report

The overall quality of the study is good, the manuscript is written (almost) clearly, and the study protocol is easy to reproduce. 

The study covers the effects of mild hypothermia on cerebral microcirculation and cerebral metabolism. One of the main questions raised is whether cerebral microcirculation can decrease accordingly as an adaptive effect, or whether it can be affected by hypothermia, causing a "mismatch" between flow and cell metabolism (Line 78 -38). 

Major issues:

1)    My impression after reading the manuscript was the lack of a clear distinction between hypothermia and rewarming. This is important for the interpretation of the results and conclusions - the study aimed to assess the effect of hypothermia on brain tissue, but part of the conclusions was drawn based on the data from rewarming: line 449 "cerebral microvascular perfusion and metabolism were reduced, mainly at rewarming."  Indeed, the lactate/pyruvate ratio decreased in the HT and HT-oxy groups, with a subsequent increase during rewarming, but did it reach statistical significance during hypothermia? The other metabolites did not differ between groups (table 5). Thus, decreased perfusion in the microcirculation was not associated with significant metabolic changes - I suggest exposing this more in the conclusions - as in lines 349-351 and 362-366. Also, the optimal rewarming rate has not been established in the literature, and your study was not designed to assess the rewarming period.

2)    Line 368 - I would avoid the term "maladaptive" for phenomena we do not fully understand. (especially with heterogeneity index p>0.05 table 4)

3)    The graphical form and the method of presenting tabular data are highly unreadable and need to be rewritten. It is difficult to assess which values have reached the level of statistical significance (both between groups and time points). Are the p-values given only for "a" and "b" and the other differences between groups and time points statistically significant/insignificant? It has to be 100% clear.

Minor issues:

1) there are some typos in the text 

2) Line 113 - please explain why you ventilated the animals with 100% O2?

3) Please unify the units - both "ml" and "mL" occur in the text (e.g., lines 126 and 132). 

4) Line 161 - ABG in pH stat? alpha stat? This may be of marginal significance for mild hypothermia but should be included in the text.  

5) Line 198 - mcm = circular miles? Unit error? Shouldn't it be micrometers? 

6) Line 359 - in the sentence "... with an of MFI after resuscitation ...", the word "decrease" is missing? 

7) Line 276 - lactate "levels" -> concentration (similarly PaO2 line 225, 276 etc.). 

8) What were the cooling and rewarming rates (1C/h?)

8) Line 199 - I suggest adding the heterogeneity index (HI) abbreviation. To make the publication more readable, I suggest briefly explaining HI and the causes and consequences of high HI. 

9) Table caption Line 273 - differences AMONG groups (=inside=between time points?). Please clarify. 

10) Line 290 - SRI -> SVRI

11) Line 326 - I don't think this is a limitation. 

12) 410 - it is worth clarifying what the LPR indicates (as was explained in the other metabolites)

13) 432 - from the description of anesthesia, it appears that the animals were given full anesthesia and not deep sedation

14) 451 - cerebral microvascular perfusion and metabolism were REDUCED at rewarming?

Author Response

REVIEWER 1

  1. The overall quality of the study is good, the manuscript is written (almost) clearly, and the study protocol is easy to reproduce. 

We thank the Reviewer for this comment.

  1. My impression after reading the manuscript was the lack of a clear distinction between hypothermia and rewarming. This is important for the interpretation of the results and conclusions - the study aimed to assess the effect of hypothermia on brain tissue, but part of the conclusions was drawn based on the data from rewarming: line 449 "cerebral microvascular perfusion and metabolism were reduced, mainly at rewarming."  Indeed, the lactate/pyruvate ratio decreased in the HT and HT-oxy groups, with a subsequent increase during rewarming, but did it reach statistical significance during hypothermia? The other metabolites did not differ between groups (table 5). Thus, decreased perfusion in the microcirculation was not associated with significant metabolic changes - I suggest exposing this more in the conclusions - as in lines 349-351 and 362-366. Also, the optimal rewarming rate has not been established in the literature, and your study was not designed to assess the rewarming period.

We are grateful to the Reviewer for this comment, we have modified the discussion and the conclusion section, accordingly; furthermore we have modified the discussion and the limitation section on rewarming weight and rate.

  1. Line 368 - I would avoid the term "maladaptive" for phenomena we do not fully understand. (especially with heterogeneity index p>0.05 table 4)

We have modified the text, accordingly.

  1. The graphical form and the method of presenting tabular data are highly unreadable and need to be rewritten. It is difficult to assess which values have reached the level of statistical significance (both between groups and time points). Are the p-values given only for "a" and "b" and the other differences between groups and time points statistically significant/insignificant? It has to be 100% clear.

We thank the Reviewer for this remark. We have decided, according also to other Reviewers’ comments to put the tables in a supplementary file in order to lighten the manuscript and to make data clearer. Figure legends have been modified, accordingly.

  1. there are some typos in the text 

We have changed them, as requested.

  1. Line 113 - please explain why you ventilated the animals with 100% O2?

The rationale of this study was to create a healthy animal model to act as a reference for a cardiac arrest one; as cardiac arrest patients and experimental animals are ventilated with FiO2 1 routinely at the beginning of resuscitation, we chose to stick to this attitude for our healthy model. We have added an explanation sentence within the method section.

  1. Please unify the units - both "ml" and "mL" occur in the text (e.g., lines 126 and 132). 

We should have adequately unified all units.

  1. Line 161 - ABG in pH stat? alpha stat? This may be of marginal significance for mild hypothermia but should be included in the text.  

We thank the Reviewer for this remark. As described in line 248 of the present version, we used temperature corrected blood gas values, but we have added this detail also at the beginning of the Method section within the text.

  1. Line 198 - mcm = circular miles? Unit error? Shouldn't it be micrometers? 

We thank the Reviewer for this remark and we corrected the typo.

  1. Line 359 - in the sentence "... with an of MFI after resuscitation ...", the word "decrease" is missing? 

We adequately corrected the missing word.

  1. Line 276 - lactate "levels" -> concentration (similarly PaO2 line 225, 276 etc.). 

We unified all the expressions, despite repetition.

  1. What were the cooling and rewarming rates (1C/h?)

We thank the Reviewer for this comment; we modified the method section so as to clearly describe both cooling and rewarming rates.

  1. Line 199 - I suggest adding the heterogeneity index (HI) abbreviation. To make the publication more readable, I suggest briefly explaining HI and the causes and consequences of high HI. 

We thank the Reviewer for this suggestion; we added a short comment on HI and its alteration consequences within the text.

  1. Table caption Line 273 - differences AMONG groups (=inside=between time points?). Please clarify. 

We are grateful to the Reviewer for this remark; we have modified the text, accordingly.

  1. Line 290 - SRI -> SVRI

We thank the Reviewer for this remark and we corrected the typo.

  1. Line 326 - I don't think this is a limitation. 

We thank the Reviewer for this comment, however we dont see a Limitation at line 326.

  1. 410 - it is worth clarifying what the LPR indicates (as was explained in the other metabolites)

We thank the Reviewer for this suggestion that has been satisfied.

  1. 432 - from the description of anesthesia, it appears that the animals were given full anesthesia and not deep sedation

We agree with the Reviewer for this comment and we modified the text, accordingly.

  1. 451 - cerebral microvascular perfusion and metabolism were REDUCED at rewarming?

We thank the Reviewer for this remark; the conclusions have been modified so as to be more coherent with both results and discussion (lines 499-501)

Reviewer 2 Report

The aim of this study was therefore to evaluate the effects of TTM on brain microcir-82 culation, oxygenation and metabolism in a healthy pig model. … whether these effects could be observed also in healthy animals remains unknown.

Conclusions: In healthy animals treated with cooling at 33-35°C, cerebral microvascular perfusion and metabolism were reduced, mainly at rewarming and potentially enhanced by con-451 comitant hyperoxia.

è But healthy hardly inform; they serve as controls

There are not enough supportive results for those key findings as stated in the first paragraph of discussion (page 12, line 347-352). No statistical results indicating those “associations” as the authors claimed between parameters were provided in this study.

The conclusions are not at all exciting or informative:

-       No baseline differences: cooling phase core/brain temperature measured

Strengths

-       15 pigs; substantive effort

-       Tremendous number of measurements

-       Arterial and mixed venous blood samples Haemoglobin oxygen saturation mean arterial pressure (MAP), pulmonary arterial pressure (PAP), right atrial 165 pressure (RAP) and pulmonary occlusion arterial pressure (PAOP) Core temperature and cardiac output, cardiac index (CI), stroke volume (SV), and systemic vascular re-169 sistances (SVR), pulmonary vascular resistance (PVR), left ventricular stroke work index 170 (LVSWI), oxygen delivery (DO2), oxygen consumption (VO2) and oxygen extraction (OER)

-       A number of vascular imaging, measurements: The mean flow index (MFI), the proportion of small-perfused vessels (PSPV) and the functional capillary density (FCD)

-       heterogeneity index for MFI, PPV and FCD,

-       Fig. 1 and Tables are normative content dump; expected differences. Commendable measurement effort; nothing new, though.

Weaknesses - Last, as TTH reduces cell metabolism, it is unknown whether systemic and brain microcirculation may ecrease accordingly as an adaptative phenomenon or if they might be affected by the 80 reduced temperature, leading to a discrepancy between flow and cell metabolism.

è Is the focus cell metabolism?

After rewarming, all microcirculatory variables returned towards normal values, except LPR which increased 3during the rewarming phase in the two groups subjected to HT when compared to the group maintained at normothermia.

è No significance is evident.

Capillary, flow differences evident…Why at 7 hours? What is the reason/explanation?

è Main conclusion is only; cortical microvascular flow is progressively reduced during cooling; not very surprising.

è “we were able to identify microcirculatory 395 abnormalities in both groups treated with hypothermia.” This is puzzling too.

è “On the other hand, we cannot exclude that BP changes might have participated to the microcirculatory alterations”…this is not explanatory

Any explanation, like “Microcirculation may play a crucial role when cellular metabolism adapts to temper-353 ature variations” is not supported by any cellular measurements.

Confusing results on declining glucose and glutamate…increasing lactate…implying deteriorating animals in both groups?

Cerebral oxygenation increased progressively during cooling only in the HT group,

-       That is surprising…what is the explanation? Any other studies confirm?

Conclusions: In healthy animals, cooling at 33-35°C can be associated with 41 alterations in the cerebral microcirculation and altered metabolism at rewarming.

è Healthy animal study with the key conclusion “associated with alteration” is not worthy of publishing: this is simply a normal response observation.

è Number of vital signs presented; temp, CI, MAP, SV, HR…VO2, SVO2, …lactate

è Key results on hypothermia and their therapeutic effect are missing.

What was the basis for selecting n=5 each group?

This study investigated the effects of cooling using target temperature management on cerebral microcirculation, oxygenation, and metabolism in a healthy pig model. Some critical questions need to be further addressed.

Methods:

-       There is no explanation about what method was used to keep the PaO2 levels constant in the HT-Oxy group.

-       Please provide details of how animals were allocated into different groups.

-       Please specify what results were analyzed by Student’s t-test and Mann-Whitney U test.

-       Please elucidate what correlations between results were interpreted by using the mixed model as indicated in the statistical analysis.

-       When using 2-way ANOVA and a significant interaction between time and group was revealed, please identify the main contributor for this result. Is it due to a significant change over time or a significant difference between groups or both?

Results:

-       The arrangement of tables, figures, and legends is very hard to follow.  Too many numbers, too many details…most not relevant to the conclusions, other that gathering ALL data just to observe.

-       Missing data in table 1, regarding PAOP.

-       abbreviations provided in the additional explanation for table 4 did not match the contents of this table.

-       figure legends were too simple.

-       no significant results between groups were indicated in the figures.

-       the p value of Heterogeneity PPV % in figure 2 (shown as p<0.001) does not match the p value of the same parameter in table 4 (shown as p=0.18).

-       There is a lack of evidence for the statement (page 11, line 324) that a higher HI for all microvascular variables was noted in HT and HT-Oxy groups compared to NT group, as no statistical significance was revealed in these comparisons.

-       According to the results in table 2, it appears that the animals were hyperventilated, especially the NT and HT group; the hemoglobin levels were relatively low in all three groups, as shown in table 3; these factors may influence the results observed in this study, please discuss.

Discussion:

-       As mentioned in the limitations, the cooling and rewarming periods were extremely short in this study when compared to clinical protocols. Also, the cooling method of using cold saline infusion in conjunction with ice packs and a nasal cooling set was extreme effective as the brain temperature was reduced to 31-33ËšC in HT and HT-Oxy group within the first hour while the core temperature remained around 35-37ËšC during the same period. It is very hard to exclude the potentially detrimental effects of speedy cooling and rewarming on the observations, which needs to be addressed before drawing any conclusion.

Other minor matters

LIne 152: You mention a fourth hole through the skull here. However, above, I only see two craniotomies. Is there another hole that is not mentioned, or should this be the third hole?

Lines 178-179: you mention administration of KCl and glucose in the case of hypokalemia or -glycemia. Were there any significant differences between groups in the frequency of need of administration? How often did these need to be administrated?

LIne 180: Was the SDF microscope removed after each set of videos in order to cover the brain with gauze and saline again? Or was it left in place? If removed and replaced, was there any method to ensure replacement in the correct position?

Lines 195-197: more description of the calculations of the parameters such as MFI, PSPV, FCD would be helpful. I see the references provided, but a small description of each parameter would be helpful in this case.

Lines 224-226: please specify the method with which you kept PAO2 constant in the HT-Oxy group. Was it through respiratory rate?

Additionally, I noticed that the caption for Table 4 provided the incorrect abbreviation definitions. Please revisit this caption and provide full names of FCD, MFI, etc, rather than the metabolic variables.

Side note: is there any better way to present these results besides huge tables? We understand that you want to show all the variables, BUT only show graphs of the ones they deemed significant to the study. Would it be better to remove the tables entirely, and graph more of the variables instead. Perhaps if a variable is does not have any interesting things to add, it can be included in the supplement instead.

Author Response

REVIEWER 2

  1. The aim of this study was therefore to evaluate the effects of TTM on brain microcirculation, oxygenation and metabolism in a healthy pig model. … whether these effects could be observed also in healthy animals remains unknown. Conclusions: In healthy animals treated with cooling at 33-35°C, cerebral microvascular perfusion and metabolism were reduced, mainly at rewarming and potentially enhanced by concomitant hyperoxia.

We thank the Reviewer for this comment; the rationale for our study was to evaluate on a healthy animal model the effects of temperature on brain dynamics so as to create a first assessment of this intervention on a healthy brain.

  1. There are not enough supportive results for those key findings as stated in the first paragraph of discussion (page 12, line 347-352). No statistical results indicating those “associations” as the authors claimed between parameters were provided in this study.

We thank the Rewier for this remark; we agree this is a descriptive study of time-course of variables and it is hard to assess specific causality. However, we have used the word “association”, which translates this uncertainty.

  1. The conclusions are not at all exciting or informative: No baseline differences: cooling phase core/brain temperature measured

We thank the Reviewer for this remark, however we don’t see which group has reported similar results on a complete neuromonitoring system in the literature before. We therefore kindly disagree with this comment.

  1. Strengths: 15 pigs; substantive effort; Tremendous number of measurements; Arterial and mixed venous blood samples Haemoglobin oxygen saturation mean arterial pressure (MAP), pulmonary arterial pressure (PAP), right atrial 165 pressure (RAP) and pulmonary occlusion arterial pressure (PAOP) Core temperature and cardiac output, cardiac index (CI), stroke volume (SV), and systemic vascular re-169 sistances (SVR), pulmonary vascular resistance (PVR), left ventricular stroke work index 170 (LVSWI), oxygen delivery (DO2), oxygen consumption (VO2) and oxygen extraction (OER); A number of vascular imaging, measurements: The mean flow index (MFI), the proportion of small-perfused vessels (PSPV) and the functional capillary density (FCD); heterogeneity index for MFI, PPV and FCD,

      We are grateful to the Reviewer for these acknowledgements.

  1. 1 and Tables are normative content dump; expected differences. Commendable measurement effort; nothing new, though. 

Again, we kindly disagree with this comment, as a similar approach to completely assess microcirculation,  oxygenation and metabolism over time has not been published before.

  1. Weaknesses - Last, as TTH reduces cell metabolism, it is unknown whether systemic and brain microcirculation may decrease accordingly as an adaptative phenomenon or if they might be affected by the reduced temperature, leading to a discrepancy between flow and cell metabolism.

We thank the Reviewer for this remark; we have discussed this point into the original version of the manuscript. Considering later increase of HI and LPR, we considered this phenomenon as maladaptive or pathological.

  1. Is the focus cell metabolism?

Our aims were to evaluate the effects of TTM on cerebral microcirculation, oxygenation and metabolism dynamics, as indicated into the manuscript.

  1. After rewarming, all microcirculatory variables returned towards normal values, except LPR which increased during the rewarming phase in the two groups subjected to HT when compared to the group maintained at normothermia. No significance is evident.

We have indicated the significance among groups in the Figure and text.

  1. Capillary, flow differences evident…Why at 7 hours? What is the reason/explanation?

Our swine animal model was the same used into a previous publication evaluating the effects of TH on microvascular and global haemodynamics on a healthy animal model (ref. 24,) so as to provide a reasonable means of comparison..

  1. Main conclusion is only; cortical microvascular flow is progressively reduced during cooling; not very surprising.

      We disagree with the reviewer. We have lighted other issues on oxygenation and metabolism, which deserve to be mentioned.

  1. we were able to identify microcirculatory abnormalities in both groups treated with hypothermia.” This is puzzling too.

      We do not understand why this is puzzling.

  1. On the other hand, we cannot exclude that BP changes might have participated to the microcirculatory alterations”…this is not explanatory

 We thank the Reviewer for this remark; our comment was made with the aim of raising one the possible mechanisms of the alterations we recorded, acknowledging the impossibility of providing a definite explanation.

  1. Any explanation, like “Microcirculation may play a crucial role when cellular metabolism adapts to temperature variations” is not supported by any cellular measurements.

 We understand this point but cerebral microdialysis is the best surrogate we have to have repeated in vivo measurements of cerebral metabolism.

  1. Confusing results on declining glucose and glutamate…increasing lactate…implying deteriorating animals in both groups?

 We have explained these points into the discussion. Only LPR seems to have differences amon groups; this value is also the one currently used in clinical practice to manipulate cerebral metabolism.

  1. Cerebral oxygenation increased progressively during cooling only in the HT group. That is surprising…what is the explanation? Any other studies confirm?

We thank the Reviewer for this comment; We have explained this is due to the concomitant increase in PaO2.

  1. Conclusions: In healthy animals, cooling at 33-35°C can be associated with alterations in the cerebral microcirculation and altered metabolism at rewarming.

Conclusions have been slightly modified, according to other proposals from another reviewer.

  1. Healthy animal study with the key conclusion “associated with alteration” is not worthy of publishing: this is simply a normal response observation. Number of vital signs presented; temp, CI, MAP, SV, HR…VO2, SVO2, …lactate. Key results on hypothermia and their therapeutic effect are missing.

 We disagree on this point. Very often healthy animals are used only as a comparison with a specific “intervention” group and not enough discussion on changes in several variables is provided. Our results will help to better understand therapeutic interventions in future studies evaluating the neuroprotectibve effects of TTM.

  1. What was the basis for selecting n=5 each group?

 The use of the minimal number of animals is actually requested by the Ethics Committee. This is a quite commonly used cohort in studies evaluating large animals with several monitoring tools.

  1. This study investigated the effects of cooling using target temperature management on cerebral microcirculation, oxygenation, and metabolism in a healthy pig model. Some critical questions need to be further addressed.

We think that the most important have been addressed. Of course, we are keen to have more input from the reviewer to further improve the manuscript.

  1. Methods: There is no explanation about what method was used to keep the PaO2 levels constant in the HT-Oxy group.

      We thank the Reviewer for this comment; as described in the method section, we modified FiO2 according to the results of blood gas analysis, hourly performed.

  1. Please provide details of how animals were allocated into different groups.

      We are grateful for this remark; animals were randomly allocated to one of the three groups by the means of blind pre-sealed envelopes; we added this detail within the Method section.

  1. Please specify what results were analyzed by Student’s t-test and Mann-Whitney U test.

      This has been specified already, it depends on the distribution of the variables.

  1. Please elucidate what correlations between results were interpreted by using the mixed model as indicated in the statistical analysis.

      We evaluated different among groups over time (and not within the same group).

  1. When using 2-way ANOVA and a significant interaction between time and group was revealed, please identify the main contributor for this result. Is it due to a significant change over time or a significant difference between groups or both?

      As reported before, we have lightlighted the differences among groups over time.

  1. Results: The arrangement of tables, figures, and legends is very hard to follow.  Too many numbers, too many details…most not relevant to the conclusions, other that gathering ALL data just to observe.

      We thank the Reviewer for this comment; we have decided to arrange all 5 tables in a Supplementary material file so as to make the whole manuscript lighter and more readable.

  1. Missing data in table 1, regarding PAOP.

      We are grateful to the Reviewer for this remark, this has been added.

  1. abbreviations provided in the additional explanation for table 4 did not match the contents of this table.

      We are grateful for this remark; Table 4 legenda has been adequately modified.

  1. figure legends were too simple.

      We thank the Reviewer for this comment; we have modified them adding the meaning of p value.

  1. no significant results between groups were indicated in the figures.

      We have added explanation for the p value, as requested.

  1. the p value of Heterogeneity PPV % in figure 2 (shown as p<0.001) does not match the p value of the same parameter in table 4 (shown as p=0.18).

      We are grateful to the Reviwer for this remark, This has been corrected.

  1. There is a lack of evidence for the statement (page 11, line 324) that a higher HI for all microvascular variables was noted in HT and HT-Oxy groups compared to NT group, as no statistical significance was revealed in these comparisons.

      We thank the Reviewer for this remark. We have modified the text, accordingly.

  1. According to the results in table 2, it appears that the animals were hyperventilated, especially the NT and HT group; the hemoglobin levels were relatively low in all three groups, as shown in table 3; these factors may influence the results observed in this study, please discuss.

      We thank the Reviewer for this remark; being the animals overall normocapnic (except for the last measurements for normothermic animals - 35±2 mmHg -, and being them all relative anemic since the baseline records, we disagree that this variables might influenced the results.

  1. Discussion: As mentioned in the limitations, the cooling and rewarming periods were extremely short in this study when compared to clinical protocols. Also, the cooling method of using cold saline infusion in conjunction with ice packs and a nasal cooling set was extreme effective as the brain temperature was reduced to 31-33ËšC in HT and HT-Oxy group within the first hour while the core temperature remained around 35-37ËšC during the same period. It is very hard to exclude the potentially detrimental effects of speedy cooling and rewarming on the observations, which needs to be addressed before drawing any conclusion. 

 We are grateful to the Reviewer for this comment. We have better addressed this aspects within the limitation part (lines 510-515)

  1. Line 152: You mention a fourth hole through the skull here. However, above, I only see two craniotomies. Is there another hole that is not mentioned, or should this be the third hole?

We thank the Reviewer for this remark. We have modified the Method section so as to be clearer about the skull management (3 skull holes but 4 dura holes). 

  1. Lines 178-179: you mention administration of KCl and glucose in the case of hypokalemia or -glycemia. Were there any significant differences between groups in the frequency of need of administration? How often did these need to be administrated?

 Unfortunately this has not been recorded.

  1. Line 180: Was the SDF microscope removed after each set of videos in order to cover the brain with gauze and saline again? Or was it left in place? If removed and replaced, was there any method to ensure replacement in the correct position?

 We thank the Reviewer for this comment; we added an explanation sentence within the Method section so as to clarify the procedure. This is the same approach used for all animals and in previous publications from the same group.

  1. Lines 195-197: more description of the calculations of the parameters such as MFI, PSPV, FCD would be helpful. I see the references provided, but a small description of each parameter would be helpful in this case.

 We are grateful to the Reviewer for this comment. We have added a short description of the cited variables within the Method section.

  1. Lines 224-226: please specify the method with which you kept PAO2 constant in the HT-Oxy group. Was it through respiratory rate?

We thank the Reviewer for this comment; we have modified the method section so as to make our management clearer. 

  1. Additionally, I noticed that the caption for Table 4 provided the incorrect abbreviation definitions. Please revisit this caption and provide full names of FCD, MFI, etc, rather than the metabolic variables.

 We are grateful to the Reviewer for this remark and we have modified Table 4 legenda adequately.

  1. Side note: is there any better way to present these results besides huge tables? We understand that you want to show all the variables, BUT only show graphs of the ones they deemed significant to the study. Would it be better to remove the tables entirely, and graph more of the variables instead. Perhaps if a variable is does not have any interesting things to add, it can be included in the supplement instead.

We thank the Reviewer for this comment and we have thus added a Supplementary file with all tables.

Reviewer 3 Report

It`s an interesting and big topic on 'The effects of cooling on brain microcirculation, oxygenation, and metabolism', but the data in this study might not meet all the topics and data is terrible all with tables and line graphs. There are four points that need to be concerned:

(1) What is your special innovation? There are many studies about hypothermia for the brain on rats (PMID: 26485658), dogs(PMID: 8701110), and porcine models (PMID: 28405859) ;

(2) Why did you choose the pigs model? Please explain the reason you chose this model and the pig`s body type.

(3) The length of time for this study. This study only focused on 24h, how about for a longer time?

(4) The terrible presence of the data. There are only tables and line graphs in the results, how about the dynamic blood flow condition with color?  And the results are only the phenomena, and what are the molecular or protein mechanism?

Author Response

REVIEWER 3

  1. It`s an interesting and big topic on 'The effects of cooling on brain microcirculation, oxygenation, and metabolism', but the data in this study might not meet all the topics and data is terrible all with tables and line graphs.

We thank the Reviewer for this comment; we have added a Supplementary file with all tables so as to make the whole manuscript lighter and more readable.

  1. What is your special innovation? There are many studies about hypothermia for the brain on rats (PMID: 26485658), dogs(PMID: 8701110), and porcine models (PMID: 28405859) ;

We are grateful to the Reviewer for this question. First, small animal models might not provide adequate translational information. Second, as nicely reported by previous works, cerebral dynamics after cardiac arrest are altered during hypothermia, nevertheless, lacking of a healthy model, it is not possible to definite identify the impact of the various interventions.

  1. Why did you choose the pigs model? Please explain the reason you chose this model and the pig`s body type.

We thank the Reviewer for this question; we chose a pig model as cardiovascular anatomy is similar to human ones and brain circulation has been largely studied for microcirculation. Also, as we planned to evaluate the same intervention in a cardiac arrest model, pig is the preferred species for this.

  1. The length of time for this study. This study only focused on 24h, how about for a longer time?

We are grateful to the Reviewer for this comment; as described within the text, we chose our timeline in agreement with other studies already performed by our group; furthermore, longer study periods might have implied the need for additional therapies such as prophylactic/therapeutic antimicrobials, etc…

  1. The terrible presence of the data. There are only tables and line graphs in the results, how about the dynamic blood flow condition with color?  And the results are only the phenomena, and what are the molecular or protein mechanism?

We thank the Reviewer for this comment; as replied above, we have added a Supplementary file with all tables so as to make the whole manuscript lighter and more readable. As far as the other two raised aspects, the lack of mechanistic explanations is one of our Limitations.